# Adoption of Fintech Services in Young Students: Empirical Approach from a Developing Country

**María Camila Bermeo-Giraldo** [1] , **Alejandro Valencia-Arias** [2,*] , **Lucia Palacios-Moya** [3] **and Jackeline Valencia** [4]

1 Facultad de Ciencias Económicas y Administrativas, Instituto Tecnológico Metropolitano, Medellín 050034, Colombia; mariabermeo@itm.edu.co

2 Escuela de Ingeniería Industrial, Universidad Señor de Sipán, Chiclayo 14001, Peru

3 Centro de Investigaciones Escolme-CIES, Institución Universitaria Escolme, Medellín 050005, Colombia; ciessalud3@escolme.edu.co

4 Instituto de Investigación y Estudios de la Mujer, Universidad Ricardo Palma, Lima 15039, Peru; javalenca.a@gmail.com

* Correspondence: valenciajho@crece.uss.edu.pe; Tel.: +57-3002567977

**Abstract:** This work aimed to identify the main variables that determine the adoption of Fintech services in young students in the Colombian context through a model with five factors proposed to explain this behavior in 124 Colombian university students. The methodological design followed a quantitative approach and an exploratory–descriptive scope. For data processing, the statistical techniques exploratory factor analysis (EFA) and confirmatory factor analysis (CFA) were used to extract the relevant factors and evaluate the measurement model. To test the hypotheses about the relationships of the conceptual model constructs, Cramer's V coefficient was used. The results showed that financial education and social influence have a positive effect on perceived benefit; in turn, low regulation is not strongly related to perceived benefit and is not dependent on social influence. However, digital literacy is affected by financial education and social influence. It is concluded that the number of mobile users in Colombia is increasing rapidly; however, the adoption of Fintech is slow. In addition, most of the university students in this study do not know what Fintech is, but they recognize that they use it frequently.

**Keywords:** university students; Fintech; adoption models; Colombia

## 1. Introduction

Technology in general, and information and communication technologies (ICT) specifically, have directly impacted all facets of human life, from innovation processes that affect the economy and industrial and organizational dynamics to important advances in different sectors. Such is the case in the financial sector, where emerging disruptive technologies such as financial technologies (Fintech) are adding elements of ease and speed to the different transactions carried out in that sector (Setiawan et al. 2021).

Likewise, in the words of the authors (Candra et al. 2020), the Fintech concept, which combines the words "financial" and "technology", accounts for a high degree of innovation in the field of financial services, adding technological elements to financial activities, which gives rise to a new sector within the financial industry, where commerce, corporate business and consumer services are supported.

In this sense, the increasingly widespread and accelerated disruption of Industry 4.0 technologies has notably revolutionized the financial services industry, also expanding the discussion around the generation of competitive advantage to survive in globalized markets (Abdul-Rahim et al. 2022). Therefore, with the increasingly complex integration of internet technologies into the financial industry, Fintech has been able to offer a series of innovative financial services, such as online payments, peer-to-peer loans, budgeting, crowdfunding and savings and investments (Xie et al. 2021).

In this sense, in the scientific literature, two types of research associated with Fintech are identified: the first is about its revolution and effects in an established financial industry, where the understanding of the mechanisms of the Fintech platforms themselves are discussed, and the second is associated with the investigation of the factors that affect the adoption of Fintech platforms (Xie et al. 2021), which, according to Hasan et al. (2021), are due to their proven ability to reduce investment barriers, as well as their multiple benefits and facilities.

It is worth mentioning that the adoption of Fintech platforms is also heavily influenced by the current historical context, specifically the period after the COVID-19 pandemic, during which, due to public health issues, social distancing increased and, with this, the need to adopt technologies for the provision of various services, including financial services, arose (Xie et al. 2021). It is for this reason that other authors such as Fu and Mishra (2022) discuss the importance of Fintech today, understanding that the adoption of digital finance has contributed significantly to households and companies, recognizing that this adoption, which has taken off in an accelerated and massive manner, also has important implications for balancing the market between traditional holders and Fintech-based new users.

On the other hand, it is important to mention that individuals' adoption behavior regarding Fintech manifests both in the behavior of adopting technology itself and in the behavior of consuming financial services (Xie et al. 2021), which is why, through the use of smartphones or other mobile devices, customers, payment providers and merchants can be connected to complete transactions (Hasan et al. 2021).

In the field of fintech adoption research, several recent studies have contributed to the literature by addressing various aspects and generating new insights in the application of fintech services. For instance, Daragmeh et al. (2022) conducted a study on the anticipated utilization of authentication models in the post-wallet period. Similarly, Yan et al. (2021) examined the factors influencing the readiness to use mobile financial services, with a specific focus on the role of fintech during the COVID-19 pandemic. Other studies have concentrated on specific areas, such as the analysis of next-generation IoT in the fintech ecosystem by Maiti and Ghosh (2023), and the investigation of the impact of development finance on open innovation in emerging economies by Mikhaylov et al. (2023).

Additionally, Najaf et al. (2022) compared the sustainability and capital performance of fintech and non-fintech firms, while Chen et al. (2023) explored the influence of gender imbalance in the population on fintech innovation. On the other hand, Ren et al. (2021) proposed an adaptive recovery mechanism for SDN controllers in cloud-based fintech applications, and Jinasena et al. (2020) reviewed previous perspectives on the success and failure of fintech projects, emphasizing the significance of project management in the field. Some studies have focused on user experience and psychology, such as Jangir et al. (2022) investigating the mitigating effect of perceived risk on the retention intention of fintech service users. Additionally, Sun et al. (2023) assessed the current state of fintech research, while Festa et al. (2022) conducted research on the impact of fintech determinants on entrepreneurship in Tunisia.

From another perspective, the adoption of Fintech is vitally associated with various current political approaches which have arisen through the Sustainable Development Goals (SDGs), which discuss the importance of adopting clean energy (Fu and Mishra 2022). This is supported by Abdul-Rahim et al. (2022), who indicate that Fintech, as a process derived from Industry 4.0 technologies, allows optimizing efficiencies as an important agent of change for sustainability.

In this sense, based on the importance of the adoption of Fintech for users, companies and economic sectors in general, various authors have tried to study it based on the validation of the main theories available in the scientific literature. Authors such as Setiawan et al. (2021) apply theories such as the technological acceptance model (TAM), technological readiness (TR), interpersonal behavior theory (IBT), as well as the unified theory of acceptance and use of technology (UTAUT) that, in addition, were validated by

authors such as Singh et al. (2020) studying variables or constructs from three aspects: adoption, behavior and the technology itself.

On the one hand, access to financial services is still considered one of the main problems faced by communities around the world (Yan et al. 2021), and on the other hand, the adoption of Fintech lags behind among consumers, especially in developing countries and emerging markets (Mazambani and Mutambara 2020; Hasan et al. 2021).

Considering the above, financial education for countries with emerging or developing economies, such as Colombia, is significantly influenced by levels of education (Karakurum-Ozdemir et al. 2019), with education also being a fundamental tool for the implementation of ICT, which in turn precedes the discussion on digital literacy (Díaz Vásquez and Bejarano Pérez 2021). Therefore, the university population is considered a determinant for understanding the factors that determine digital literacy, as a variable, and in turn, a determinant in the discussion on adopting Fintech.

Despite the extensive literature on adoption studies, there remains a need to investigate the factors influencing consumers' intention to use FinTech services, which are still considered a novel service in the financial industry (Daragmeh et al. 2022), particularly in countries with emerging economies, such as Colombia. Promoting financial inclusion using technological innovations in these nations enhances their economic development and consequently improves people's quality of life (Apostu et al. 2023). Similarly, it is necessary to examine the financial behavior of students, who represent the future professionals comprising the workforce (Boolaky et al. 2021). Thus, their financial decisions will have a significant impact on the future economy of each country. Therefore, providing them with proper financial literacy to enhance their access to financial services and understanding how technology influences their decisions becomes essential (Dzokoto et al. 2023). In this sense, the objective of this research article is to identify the main variables determining the adoption of Fintech services among young students in Colombia, a developing country.

This document is structured as follows: After the introduction, the first section describes the literature review that presents the conceptual model and hypotheses. The second section presents the methodology used. The third section presents and interprets the results. The fourth section discusses the findings, the added value, the practical implications, and the limitations of the article. Finally, the fifth section presents the study's conclusions.

## 2. Literature Review

The term fintech encompasses the use of technology to provide financial services (Bhaskaran 2021). In recent years, there has been a growing interest in its adoption, as information and communication technologies have provided a platform for its introduction and dissemination in the market (Kanga et al. 2021), making it one of the most prominent innovations in the industry due to its transformative capacity (Iman 2020).

This rapid expansion has been driven by technological advancements and the recent global financial crisis (Firmansyah et al. 2023). These innovations have significant potential to bring about profound changes in all areas of financial services and have a significant impact on the global economy (Al_Kasasbeh et al. 2023), promoting financial inclusion, which refers to the access of underserved households and small businesses to financial products and services. This has enabled many individuals to carry out transactions according to their specific needs, even if they have not been served by traditional financial institutions (Hajin and Jajuli 2022).

These technologies have opened new opportunities for accessing financial services more conveniently, efficiently, and affordably. Companies in this sector have been able to streamline processes, reduce costs, and improve the user experience by technologies such as artificial intelligence, machine learning, and blockchain (Najaf et al. 2020; Lobozynska et al. 2023).

Therefore, to enrich and strengthen the development of hypotheses within the framework of this research, it is necessary to identify the main factors related to fintech adoption. This theoretical approach will allow for a comprehensive analysis and combination of

variables to establish a solid foundation that supports the formulation and evaluation of the hypotheses proposed in this study. In this way, a comprehensive orientation is sought to contribute to the understanding of the adoption of financial technologies, thereby promoting the obtainment of reliable and significant results.

Based on the study conducted by Setiawan et al. (2021), which demonstrates the use of variables such as social influence, perceived usefulness, low regulation, financial education, and digital literacy, it is necessary to understand these concepts according to the studies that have been carried out in this regard. According to Junnonyang (2021), social influence can be defined as the degree to which a person adopts and uses certain systems, innovations, or technologies due to the recommendation of someone close to them, as the opinions of individuals close to them, such as family and friends, have a notable influence on the predisposition to adopt digital financial services (Patel and Patel 2018). On the other hand, digital literacy refers to a set of skills necessary to use software, applications, or a digital device, which includes skills related to information evaluation and knowledge acquisition (Chetty et al. 2018), including cognitive, emotional, and sociological aspects that technology users need to effectively navigate digital environments (Eshet 2004).

Collectively, the study by Abima et al. (2021) demonstrated a significant positive relationship between social influence, digital literacy, and the intention to adopt digital media. Thus, it can be affirmed that interactions with technology and experiences of a person close to a potential user impact and lead to a better understanding of devices and improvement of their technological skills (Marsh 2021). Based on this, the following hypothesis is proposed:

**Hypothesis 1.** *Social influence has a causal effect on digital literacy.*

Furthermore, constantly evolving technological advancements also require continuous education on electronic and digital operations. According to Srirahayu et al. (2022), digital literacy greatly affects young individuals intending to adopt new financial technologies. Perceived usefulness, validated in various contexts, is understood in this study as students' interest in using a technology if they find it beneficial (Davis 1989). Additionally, as expressed by Prastiawan et al. (2021), customers choose to use financial products and services that have been tested and recommended by someone in their social circle, demonstrating that social influence has both direct effects on the use of digital financial services and direct effects on the intention to use these technologies. Based on the above, the following hypothesis is defined:

**Hypothesis 2.** *Social influence has a causal effect on perceived usefulness.*

On the other hand, a key factor among young students concerned about the use of banking products and online transactions is the risk associated with loosely regulated services (Magnuson 2018). Currently, financial systems and institutions face challenges in minimizing the risk associated with Fintech transactions due to the constant advancement of technology, which also evolves into new forms of fraud (Traif et al. 2020). Therefore, this implies consumers' perception of being at risk of information loss or theft and fraud in some Fintech transactions, as well as an association with the lack of integrity, privacy, and authenticity of information (Leong et al. 2017). In the study by Bromberg et al. (2017), students conduct an appropriate evaluation to find utility and functionality in technology-mediated financial products and services, and if they do not identify associated risks, they use them. In other words, young students will utilize Fintech services if they feel supported by regulation and if their use enhances their management (López Maldonado and Valdés Godínes 2020). Therefore, the following hypothesis is proposed:

**Hypothesis 3.** *Low regulation influences perceived benefits.*

According to Thomas and Subhashree (2020), financial education is vital for university students because knowing how to manage and utilize financial resources determines the proper functioning of any business, and students are considered future entrepreneurs. In this way, adequate financial literacy plays an important role in reducing financial vulnerability as it instills confidence among students in decision-making and entrepreneurial initiatives (Chhatwani and Mishra 2021). Regarding digital education or literacy, understanding ICT is crucial as it enables young individuals to perform all electronic banking transactions (Sentosa et al. 2022). Previous studies, such as Nazah et al. (2022), have already validated the relationship between financial and digital education with the intention to use financial technologies among student populations.

Recent studies by Lyons and Kass-Hanna (2021) and Koskelainen et al. (2023) have highlighted the link between financial literacy and digital literacy and the need to review and expand the traditional conception of financial literacy while incorporating digital literacy. Financial education can influence digital literacy by providing knowledge about FinTech tools because, according to Golden and Cordie (2022), this knowledge can promote individuals' security in the responsible and correct use of these services. Additionally, with the financial information possessed by students, they can better navigate digital applications for accessing banking services and increase their knowledge of computer operations or the latest technologies.

**Hypothesis 4.** *Financial education influences digital literacy.*

Furthermore, according to Thomas and Subhashree (2020), university students lack knowledge of basic financial terms, such as simple and compound interest estimation, loan application, inflation, budgeting, and loan alternatives. This is critical because having strong financial habits prepares individuals to face macroeconomic and unforeseen financial problems. In addition to the above, knowledge of digital banking products and services is necessary for students to easily manage their transactions (Munari and Susanti 2021). Moreover, the knowledge gained during university education remains with them when managing their personal finances and in their professional lives (Thomas and Subhashree 2020). For this study, the perceived usefulness factor in Fintech services by university students refers to the benefits obtained by these services or the satisfaction of their needs in the field of digital financial transactions. Thus, financial knowledge enhances the development and performance in the use of digital banking channels (Sentosa et al. 2022). Therefore, the greater the perceived benefit, the more competent the student will be in using Fintech financial services, as they will find it easier to carry out all banking transactions without significant difficulties (Munari and Susanti 2021). Based on this, the following hypothesis is proposed:

**Hypothesis 5.** *Financial education influences perceived benefit.*

However, the accelerated growth of Fintech also poses regulatory and security challenges. Governments and financial authorities worldwide are working to establish appropriate regulatory frameworks that promote innovation and protect consumer interests. Therefore, Fintech regulators must consider continuous updates to their legal and regulatory framework to provide a trustworthy environment free from risky transactions, as there is an increase in financial risks arising from new financial services that users are not familiar with, such as loan credits replacing traditional financial intermediation (Wronka 2023). Hence, in Foley et al.'s (2019) study, it is stated that regulators of these services have directed their concerns towards the lack of knowledge among young users and the limited supervision of services like cryptocurrencies. Efforts have been made in communication processes to disseminate the functioning of these products through social media. In other words, having regulation in Fintech services and products can influence the behavior of young students regarding their intention to use, as consumers rely on the opinions or recommendations of their personal networks about the regulations supporting the financial

system (Morales and Landeo 2021). Considering the above, the following hypothesis is proposed:

**Hypothesis 6.** *Low regulation has a causal effect on social influence.*

Considering the above, Figure 1 summarizes and presents the proposed hypotheses that support the conceptual model.

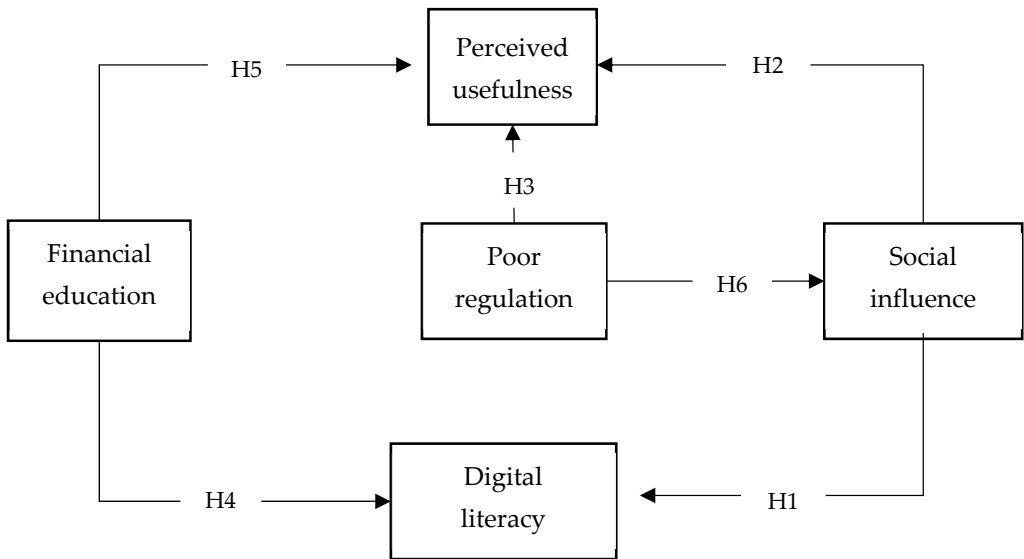

**Figure 1.** Proposed research model.

## 3. Methodology

### 3.1. Population, Sample and Collection of Information

To understand the adoption of Fintech services in young students from an empirical approach in a developing country, this study is part of a cross-sectional investigation with a correlational and descriptive scope. The population being studied is made up of young students at universities in Medellín, Colombia. The literature has validated the affinity and compatibility of the young generations with the use of financial technologies (Apostu et al. 2023) because according to Nourallah (2023), young people possess digital skills that other generations do not. Aligned with the above, in accordance with what is stated above (Apostu et al. 2023), the youth population and students access services such as purchases in mobile applications, online bill payment, bank account access and online finance management, among others. Considering the above, the sample corresponded to 124 people and was selected through convenience sampling according to the young student volunteers who wanted to participate in the study. It is worth noting that this study employed a non-probabilistic and purposive sampling approach, following Ruiz's (2008) method of selecting participants who were accessible and met the researchers' criteria, in this case, being students and users of FinTech services. Additionally, the sample is characterized by being small and heterogeneous. According to Etikan (2017), there is no minimum acceptable sample size as it can vary depending on the study's context and available resources. Moreover, the sample is not representative, which implies that the observed results cannot be generalized. However, these samples are valid and useful for specific research objectives and can help answer particular research questions and generate new hypotheses (Stratton 2023). Hence, the inclusion of 124 participants is considered an adequate number to obtain significant results on financial behavior and technological adoption in an exploratory and non-probabilistic manner, which can later serve as input for more robust studies. However, it should be acknowledged that the purposive sample carries inherent bias in representativeness, and this limitation should be clearly stated in

the research (Ruiz 2008), as evidenced in Section 4.4, "Limitations of this study." The survey was administered in person during the last quarter of 2022.

### 3.2. Instrument Design

To facilitate the understanding and analysis of the surveys in the database, the information collection instrument was divided into three main parts. The first part consists of the demographic characteristics of the respondent. This section has four questions in total. The second section corresponds to five questions to measure the knowledge, use and access of the respondents about financial services and products mediated by technology. The last section was based on 20 statements or items with a five-point Likert-type categorical measurement scale. The respondent had to respond between "completely disagree" (1) and "completely agree" (5). The purpose of this questionnaire was to understand with an empirical approach the adoption of Fintech services in young students in a developing country (Appendix A). In this sense, the dimensions of the model proposed in the 19 items were based on empirically validated factors in previous studies in the context of technological adoption and the use of innovations in banking services: (1) social influence, behavioral sciences and psychology of individuals indicates that people follow the behavior of their close circle (Lu et al. 2005; Patel and Patel 2018); (2) digital literacy, the disposition and level of familiarization that the person has towards technology (Choi et al. 2022; S. A. L. H. Munari and Susanti 2021); (3) low regulation, regulatory policies are those that strengthen the intention to use financial technology (Choi et al. 2022; McCallum and Aziakpono 2023); (4) financial education, which is one of the key factors in the intention to use digital financial services and products (Morgan and Trinh 2019; Pandey et al. 2022); and (5) perceived usefulness, which has been demonstrated to be a determining factor in the adoption of technology (Prastiawan et al. 2021; Liu et al. 2022). Additionally, the questionnaire was validated by two experts in the financial area (university professors), who carefully reviewed all the items before verifying that each element was developed according to the context of the research variables.

### 3.3. Data Processing

As a data analysis technique, a factor analysis (hereinafter FA) was used to analyze the perception of Fintech in the personal finances of this population through the statistical program of IBM, SPSS for Windows, version 22 (IBM Corp. 2013). FA is a statistical data reduction technique by which the interdependence of variables can be examined and provides the underlying structure of the data. Using this technique, it is possible to determine the minimum number of factors that explain the variability of the data provided by an instrument (Montilla 2013). Conceptually, FA is composed of two different types or modalities, exploratory factor analysis (hereinafter EFA) and confirmatory factor analysis (hereinafter CFA). In the first step, EFA is used to define the proposed theoretical model, and in the second step, CFA is used to validate this model (Pérez-Gil et al. 2000). In the EFA, the maximum likelihood (ML) procedure is applied to the set of variables or dimensions for the extraction of the factors and the VARIMAX procedure for their rotation. Second, the proposed model seeks to validate the relationships in the hypotheses with the factors obtained after performing the EFA, corroborating the factorial structure of standardized loads obtained for each construct through the CFA. The CFA involves estimating the convergent validity, which tests that the constructs that are expected to be related are indeed related, and it furthermore evaluates the reliability of the proposed model, using (1) the reliability of the observable items and (2) of the constructs (Calvo-Porral et al. 2013). Next, the divergent or discriminant validity is evaluated to demonstrate that the constructs that should not be correlated are not correlated (Martínez-García and Martínez-Caro 2009). Cronbach's analysis was also performed to estimate the reliability or internal consistency of the designed instrument. Finally, with Cramer's V coefficient, the hypotheses and relationships of the proposed model were tested by measuring the degree of association between the factors.

## 4. Results

### 4.1. Respondent Profile

Before analyzing the data collected as it related to the proposed model, the descriptive data on the characteristics of the respondents were examined using demographic data such as age, work, gender and educational level, as presented in detail in Table 1. The results indicate that the participants were mostly women, which is expected given that women have a high participation in higher education in the universities that were part of this study. The most common age range among respondents was 24 to 26 years, followed by the range of 18 to 20 years. Another characteristic that stood out among the respondents is that most of the young people were studying economics at a professional academic level. The sample reveals that most do not know what Fintech is; however, many have used it. Among the Fintech platforms most recognized by these young students are NEQUI and PSE. Likewise, the motivations identified by the students that influence them to use Fintech are mainly the changes in transaction time and the affinity they feel with mobile technology. It is worth noting that, as indicated in the methodology, this study followed a non-probabilistic convenience sampling. Therefore, the majority of the surveyed students were enrolled in programs related to the field of economics, as it is the faculty with the highest number of students in the institutions that were the subject of this study and corresponds to the universities where the researchers had access to collect the information.

**Table 1.** Sample information.

| Gender | Percentage | Quantity | Age range | Percentage | Quantity |
|---|---|---|---|---|---|
| Female | 72% | 89 | 18 to 20 years | 40% | 50 |
| Male | 28% | 35 | 21 to 23 years | 19% | 24 |
| | | | 24 to 26 years | 41% | 51 |
| **Program** | **Percentage** | **Quantity** | **Academic level** | **Percentage** | **Quantity** |
| Arts | 4% | 5 | Technical | 7% | 9 |
| Health sciences | 5% | 6 | | | 0 |
| Economic sciences | 44% | 55 | Technological | 17% | 21 |
| Social and human sciences | 11% | 14 | | | 0 |
| Communications | 5% | 6 | Professional | 73% | 91 |
| Law and political science | 7% | 9 | | | 0 |
| Education | 4% | 5 | Postgraduate | 3% | 4 |
| Engineering | 20% | 25 | | | 0 |
| **Do you know what Fintech is?** | **Percentage** | **Quantity** | **Have you ever used Fintech?** | **Percentage** | **Quantity** |
| Yes | 40% | 50 | Yes | 70% | 87 |
| No | 60% | 74 | No | 30% | 37 |
| **Which of the following Fintech have you heard about?** | **Percentage** | **Quantity** | **What are the factors that influence you when using Fintech** | **Percentage** | **Quantity** |
| Nequi | 65% | 81 | Lack of information | 40% | 50 |
| Payu | 4% | 5 | Uncertainty | 11% | 14 |
| PSE | 23% | 29 | Low regulation | 8% | 10 |
| Moviired | 1% | 1 | Financial education | 34% | 42 |
| None | 7% | 9 | Perceived usefulness | 7% | 9 |
| **What motivations influence young students to use Fintech?** | | | | **Percentage** | **Quantity** |
| Savings in transaction time | | | | 90% | 112 |
| Affinity with mobile technology | | | | 7% | 9 |
| The minimum cost to use these platforms | | | | 2% | 2 |
| Request fewer documents compared to a financial institution | | | | 1% | 1 |

Source: Own elaboration.

### 4.2. Factor Loads

As mentioned in the previous section, data processing was performed using EFA, which begins by analyzing the individual reliability of the items through the loads or simple

correlations of each indicator with its respective construct. These loads indicate the degree of correspondence between the variable and the factor; that is, high loads indicate that said variable is representative of said value (Montoya 2007). In this manner, the factorial loads are used to interpret the function of each variable that intervenes in the model. The most accepted rule determines a minimum threshold of 0.5, taking into account the size of the sample (Zamora et al. 2010). As shown in Table 2, most of the items exceed the minimum threshold, and the average of the factorial loads is equal to or greater than 0.7. However, in the *Low Regulation* construct, one of the items did not meet the minimum criteria and had to be eliminated.

**Table 2.** Factor load.

|  | Item | Factorial Load | Average Factorial Loads |
|---|---|---|---|
| Social influence | IS1 | 0.774 | 0.811 |
| | IS2 | 0.802 | |
| | IS3 | 0.797 | |
| | IS4 | 0.869 | |
| Digital literacy | DU1 | 0.835 | 0.761 |
| | DU2 | 0.806 | |
| | DU3 | 0.657 | |
| | DU4 | 0.745 | |
| Low regulation | BR1 | 0.701 | 0.758 |
| | BR2 | 0.783 | |
| | BR3 | 0.790 | |
| Financial education | EF1 | 0.705 | 0.661 |
| | EF2 | 0.609 | |
| | EF3 | 0.688 | |
| | EF4 | 0.641 | |
| Perceived usefulness | MO1 | 0.676 | 0.754 |
| | MO2 | 0.811 | |
| | MO3 | 0.763 | |
| | MO4 | 0.764 | |

Source: Own elaboration.

*4.3. Convergent Validity*

In accordance with what is stated in the procedure to evaluate the validity of content as explained by Luján-Tangarife and Cardona-Arias (2015), it involves applying statistical methods such as EFA, which seek to explain the existing correlations between the items of the instrument from a set of components. Therefore, in this analysis, it is crucial to evaluate the fit of the factorial model and the adequacy of the sample, and the items evaluated. In this sense, to evaluate if the set of indicators truly measures the determined construct and not another different concept, their convergence was validated using the Kaiser–Meyer–Olkin test (hereinafter KMO) and the Bartlett test of sphericity.

The KMO test is a sample adequacy test and is an index that compares the correlation coefficients with the partial correlation coefficients. In this manner, if a pair of variables are strongly correlated with the rest, the partial correlation must be small, which implies that a large part of the correlation between these variables can be explained by the other variables in the analysis. Thus, there is a strong correlation structure between them, and therefore, it makes sense to continue with FA. As a rule of thumb, KMO values less than 0.6 are considered inappropriate. Bartlett's sphericity test reflects that if there were no correlation structures between the variables involved in the FA, the correlation matrix would be an identity matrix, that is, it would have zeros off the diagonal and ones on the diagonal (Zamora et al. 2010). Table 3 shows that the result of this test was acceptable for both the KMO test and the Bartlett sphericity test, where it was found that the correlation matrix is different from the identity; therefore, it is advisable to continue with the FA.

**Table 3.** Convergent validity.

|  | Kaiser–Meyer–Olkin Measure | Bartlett's Test of Sphericity |
|---|---|---|
| Social influence | 0.773 | 0.000 |
| Digital literacy | 0.700 | 0.000 |
| Low regulation | 0.636 | 0.000 |
| Financial education | 0.674 | 0.000 |
| Perceived usefulness | 0.679 | 0.000 |

Source: Own elaboration.

*4.4. Discriminant Validity*

In this step, the degree of correlation of the items of one dimension and the score of other dimensions to which they do not belong is determined, which means that the discriminant validity will indicate that the items of each construct are not measuring what those included in the others intend to measure. To carry out this check, the range of the Pearson correlation coefficients between the questions and the domains to which they do not belong is determined to then define the percentage of success for each domain (Luján-Tangarife and Cardona-Arias 2015). To calculate the discriminant validity, this study implemented the method of comparison between the correlations of the indicators. To specify the correlations, 95% confidence intervals were constructed using the transform method (Rosnow and Rosenthal 1996). In the study of statistical tests and the evaluation of scales, the authors Martínez and Martínez in (Martínez and Martínez 2009) explain that there is discriminant validity if all the correlations between the indicators of X and Y are significant and each of these correlations is greater than all the correlations between the indicators of both variables. Table 4 shows that none of the correlations contain the value 1, which allows us to deduce that, as the constructs are not perfectly correlated, each one of them represents a different concept.

**Table 4.** Discriminant validity.

|  | IF | DU | BR | EF | MO |
|---|---|---|---|---|---|
| IF | . . . |  |  |  |  |
| DU | [0.16; 0.50] | . . . |  |  |  |
| BR | [−0.13; 0.22] | [−0.08; 0.30] | . . . |  |  |
| EF | [0.12; 0.47] | [0.17; 0.52] | [0.13; 0.46] | . . . |  |
| MO | [0.23; 0.55] | [0.05; 0.41] | [−0.08; 0.29] | [0.41; 0.67] | . . . |

Source: Own elaboration.

*4.5. Reliability*

Reliability refers to the degree to which an instrument can measure without error, and it is conceived as the internal consistency or stability of the measurements as the measuring process is repeated (Rodríguez-Rodríguez and Reguant-Álvarez 2020). One of the most used statistical resources to evaluate the reliability of an instrument is Cronbach's alpha coefficient. This statistic obtains values in the range between 0 and 1, which are closely related to the number of items that make up the scale as a mean correlation between them, which means that the more items a scale contains, the better its composition. According to the literature, the minimum accepted value for this coefficient is 0.7, since values lower than this indicate that the reliability of the scale is low. The ideal values of this coefficient are between 0.8 and 0.9, since values higher than 0.9 could indicate redundancy or duplication (Luján-Tangarife and Cardona-Arias 2015). Table 5 shows the levels of reliability that have been obtained for the different constructs that make up the Fintech model. It is observed that most of these are within the ideal range.

**Table 5.** Reliability.

| Factor | Cronbach's Alpha |
|---|---|
| Social influence | 0.879 |
| Digital literacy | 0.840 |
| Low regulation | 0.805 |
| Financial education | 0.726 |
| Perceived usefulness | 0.828 |

Source: Own elaboration.

*4.6. Hypothesis Testing*

Once the reliability and validity of the model have been evaluated, the association hypotheses are tested. There are association measures that indicate the degree of associative relationship that the variables have to determine the size of the association. This study applied Cramer's V association index. Cramer's V measure is used in a nominal contingency table, and its values are between 0 and 1. Values close to 0 indicate little association, while values close to 1 indicate strong association; however, it is very common to obtain values close to 0; therefore, the commonly accepted values are those equal to or greater than 0.3 (López-Roldán and Fachelli 2015). Table 6 presents the hypotheses of the Fintech model that will allow us to determine if the variables are significantly related and to replicate the model. The results suggest that there is a statistically significant dependence between the variables in most of the four Hypotheses.

**Table 6.** Hypothesis testing.

| | Hypothesis | Cramer's V |
|---|---|---|
| H1 | Social influence has a causal effect on digital literacy | 0.329 |
| H2 | Social influence has a causal effect on perceived usefulness | 0.392 |
| H3 | Low regulation influences perceived benefit | 0.252 |
| H4 | Financial education influences digital literacy | 0.328 |
| H5 | Financial education influences perceived benefit | 0.398 |
| H6 | Low regulation has a causal effect on social influence | 0.213 |

Source: Own elaboration.

The degree of relationship of the variables meets the criteria of the measurement indices, resulting in the relationship between financial education and perceived usefulness (0.398) being the most significant, followed by social influence with perceived usefulness (0.392). To better appreciate the degree of relationship of the variables, Figure 2 shows the influence of the constructions of the Fintech model, which allows us to discuss the results. The perceived usefulness of the use of Fintech by young university students is mainly influenced by financial education and social influence. It should be noted that low regulation (lack of standardization within the system, guarantees and regulations and cybersecurity) is not strongly related to the perceived usefulness of use and is not dependent on social influence. However, ignorance of use or lack of digital literacy (ignorance of virtual markets, regulations and trusts) is affected by financial education (0.328) and social influence (0.398).

The results previously described are observed in Figure 2.

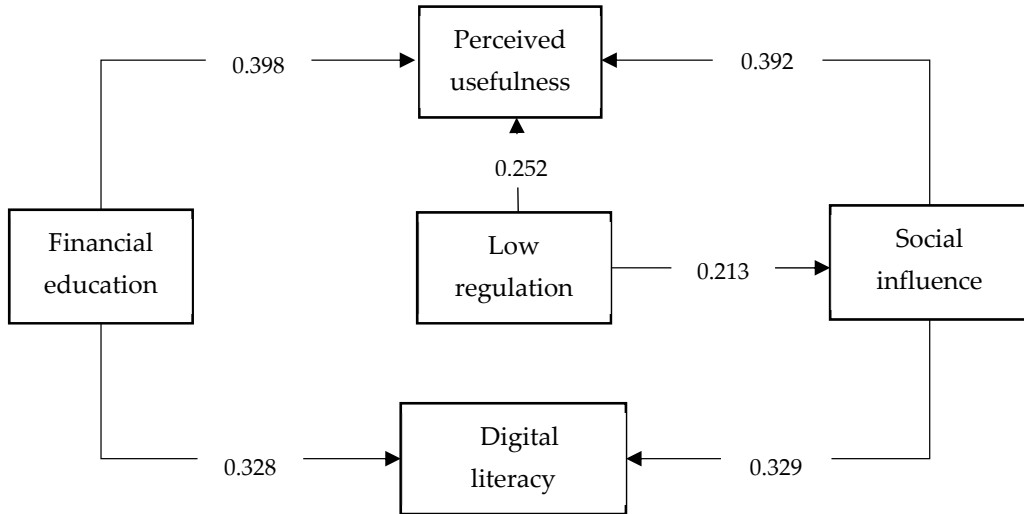

**Figure 2.** Results of the hypothesis tests.

## 5. Discussion

In the analysis of the main findings of this research, a comparative structure is proposed with the results of other similar investigations to validate variables that determine the adoption of Fintech services. This study provides additional knowledge by using theoretical adoption models in a new population that reflects the characteristics and needs of an emerging economy, such as Colombia. Like the case of Peru, where studies have investigated innovation in Fintech, finding that there are still few empirical studies on this topic (Barchi et al. 2019), which supports the literature gap on this subject in developing countries.

While the presented study provides relevant information about the characteristics of the respondents and their motivations to use Fintech in a specific context, it is important to consider the findings of other studies that have addressed Fintech adoption from different theoretical approaches and in diverse contexts. These studies highlight the influence of factors such as perceived usefulness, social influence, perceived ease of use, perceived trust, and security in the adoption of Fintech services, which can enrich the understanding of the determinants of Fintech adoption and provide a basis for future research in the field as observed in Brika's (2022) research, which has been focusing on the study of Fintech since 2018, showing that publications addressing the topic using adoption models or psychometric models have focused on elements associated with the intention to use or the adoption itself.

This assertion is corroborated in the research by Singh et al. (2020), who analyzed the factors influencing the intention to adopt Fintech among Internet users, determining the direct and positive impact of latent factors such as perceived usefulness and the direct but negative impact of social influence. Actual usage is determined by perceived ease of use and social influence, further mentioning that the intention to adopt, for the context of these internet users, is not a determining factor of adoption itself. These results, in this sense, are important for understanding the impact of factors such as social influence, which, for this research, was identified as a relevant variable for understanding both perceived usefulness and digital literacy.

In this regard, several studies have used different theoretical models to investigate the adoption of Fintech services in different contexts. For example, Mazambani and Mutambara (2020) applied the TPB in Cape Town, South Africa, to predict the intention to adopt cryptocurrencies in financial innovation adoption. They found that perceived attitude and behavior control were important factors in predicting the intention to adopt Fintech in this population. Meanwhile, Shaikh et al. (2020) used the TAM in Malaysia to examine the determinants of acceptance of Islamic Fintech services. They found that perceived ease of use, perceived usefulness, and consumer innovation were key factors in

the acceptance of these services. Additionally, Yan et al. (2021) validated the acceptance and usage of Fintech services using the UTAUT model in Bangladesh. They found that social influence, perceived value, and perceived trust were important factors in understanding the intention to use the technology.

Furthermore, studies such as those by Ali et al. (2021) and Wang (2021) have highlighted the importance of variables related to privacy, security, and perceived trust in the adoption of Fintech services. These findings provide an important contribution to the theoretical development of future adoption models that have not been addressed in the present study. Overall, the combination of these studies demonstrates the diversity of theoretical models used and the different factors considered to understand the adoption of Fintech services in different populations and contexts.

According to the above, this study positions itself as one of the few that analyze the behavior of internal latent variables such as financial education, low regulation, social influence, perceived usefulness, and digital literacy. These last two variables are important for the analysis of the intention to adopt, adoption itself and the acceptance of Fintech services.

Another added value of the research is the contribution of a new theoretical model that integrates some variables extracted from the main theories available in the scientific literature, such as TAM and UTAUT, as two of the main theoretical models regarding technology acceptance, enriching the existing knowledge about the adoption of Fintech services.

In addition to its theoretical contribution, the research also has significant practical implications. The importance of using predictive models to understand consumer behavior in a specific population, in this case, Colombia, is emphasized. By better understanding the preferences and needs of Colombian consumers, the adoption of Fintech services can be promoted within the broader context of financial innovations.

Therefore, the implications of this research encompass various areas, including policies, practices, and management. From a policy perspective, the findings provide a basis for Colombian authorities to promote the adoption of Fintech services among young students through strategies that foster financial education and raise awareness about the benefits of these innovations.

From a practical perspective, organizational leaders can utilize these findings to design strategies that drive behavioral change among Fintech users by developing intuitive interfaces tailored to user needs. Additionally, these theoretical implications can serve as a foundation for future research, enabling a deeper understanding of the internal variables that influence the adoption of Fintech services.

Regarding management, Fintech industry professionals can leverage these results to improve their business strategies, emphasizing perceived usefulness and social influence as key factors to promote the adoption of Fintech services while complying with relevant regulations to ensure user trust, security, and the need for regulations, such as Mexico's Fintech Law (Gonzalez et al. 2020).

However, it is important to emphasize that this research has two significant limitations. The first limitation is that, due to the data being collected in the context of young university students in Colombia and based on a small sample selected through a non-probabilistic and intentional method by the researchers, this study may have associated biases. Therefore, direct generalizations and inferences cannot be made to other populations without first understanding these dimensions in relation to their characteristics and needs.

On the other hand, this research was conducted within a specific period, which hinders generalization within the same Colombian population for another historical moment, as temporal changes were not considered. It is understood, primarily, that financial technologies or Fintech change rapidly.

Despite these limitations, this research provides valuable contributions to the knowledge about the adoption of Fintech services in Colombia. The findings and practical implications can serve as a starting point for future research that addresses these limitations and expands the understanding of internal factors influencing the adoption of Fintech services in different contexts and historical periods.

## 6. Conclusions

This research identified the main variables that determine the adoption of Fintech services in young students from a developing country, Colombia, through five proposed factors, considering previous studies in the context of financial services and to better understand how consumers in an emerging economy interact and engage with technology to use financial services.

The results of this study empirically and theoretically support the proposed model and provide valuable information on the factors that contribute to understanding student Fintech users. First, financial education and social influence have a positive effect on the perceived usefulness of using Fintech by young university students; therefore, the intention of deciding to use these services is determined by the benefits that they first identify from technology. Second, for students, regulations are not relevant, nor are the opinions of their close social circle regarding Fintech relevant to whether they think that it is useful to them since poor regulation does not have any causal effect on the perceived benefit of use or on social influence. Finally, university students perceive that digital literacy or knowing how financial technologies function is related to having received financial education and the social influence of their friends and family.

In addition to the above and according to the opinions of the students surveyed, consumers and users of Fintech services are clear about the benefits and the perception of usefulness represented by using them, which can be attributed to other dimensions that are worth exploring in future studies, such as impact due to COVID-19 contingency planning and the level of technological innovation to access these financial services.

The results showed that the number of mobile users in Colombia is increasing rapidly; however, the adoption of Fintech is slow. In addition, most of the university students in this study do not know what Fintech is or do not recognize it by this name, but they do use it frequently.

**Author Contributions:** Conceptualization: M.C.B.-G. and A.V.-A.; Methodology: J.V. and A.V.-A.; software: A.V.-A. and M.C.B.-G.; Validation: J.V. and L.P.-M.; Formal analysis: M.C.B.-G. and A.V.-A.; Resources: M.C.B.-G., A.V.-A., J.V. and L.P.-M.; Data Curation: A.V.-A.; Writing—original draft preparation: M.C.B.-G., A.V.-A., J.V. and L.P.-M.; Review and Editing: J.V. and L.P.-M.; Project Administration: M.C.B.-G.; Supervision: L.P.-M. All authors have read and agreed to the published version of the manuscript.

**Funding:** This research received no external funding.

**Informed Consent Statement:** Informed consent was obtained from all subjects involved in the study.

**Data Availability Statement:** Data will be provided on request.

**Conflicts of Interest:** The authors declare no conflict of interest.

## Appendix A

**Table A1.** Constructs and proposed items.

| | | | | | | | | | | |
|---|---|---|---|---|---|---|---|---|---|---|
| The purpose of this survey is for academic purposes, aiming to identify the main variables determining the adoption of Fintech services among young students in a developing country, Colombia. | | | | | | | | | | |
| 1. | Please indicate your gender | | | | | | Male | | Female | |
| 2. | Select your age range | | | | | | | | | |
| a. | 18 to 20 years old | b. | 21 to 23 years old | c. | 24 to 26 years old | | | | | |
| 3. | Select the Program area to which you belong | | | | | | | | | |
| a. | Arts | b. | Health Sciences | c. | Economic Sciences | d. | Social and Human Sciences | e. | Communications | |
| f. | Law and Political Science | g. | Education | h. | Engineering | | | | | |
| 4 | Indicate Academic level of your program | | | | | | | | | |
| a. | Technician | b. | Technological | c. | Professional | d. | Postgraduate | | | |
| 5 | Do you know what Fintech is?? | | | | | | | | | |
| a. | Yes | | | | b. | | No | | | |
| 6 | Considering that Fintechs are companies that provide financial products and services through technology, applications, and other computer-based means, have you ever used Fintech? | | | | | | | | | |
| a. | Yes | | | | b. | | No | | | |
| 7 | Which of the following Fintechs have you heard of? | | | | | | | | | |
| a. | Nequi | b. | Payu | c. | Pse | d. | Moviired | e. | No | |
| 8 | What factors influence your decision to use Fintech? (You can choose one or more) | | | | | | | | | |
| a. | Absence of information | b. | Uncertainty | c. | Low regulation | d. | Financial education | e. | Motivation | |
| 9 | What are the motivations of young students that influence their decision to use Fintech? (You can choose one or more) | | | | | | | | | |
| a. | Savings in transaction time | b. | Affinity with mobile technology | c. | The minimum cost to use these platforms | | d. | They request few documents compared to a financial institution to apply for a loan | | |
| 10 | Select with an X in the following statements the level of agreement or disagreement you have with each of them | | | | | | | | | |

| Statements | Totally agree | Agree | Neither agree nor disagree | Disagree | Totally disagree |
|---|---|---|---|---|---|
| I have a relative or friend who uses Fintech | | | | | |
| Someone recommended that I use Fintech | | | | | |
| According to other people's opinions, these platforms make the processes more agile | | | | | |
| Taking these questions into account, I would be willing to use Fintech | | | | | |
| Digital ignorance discourages investing in Fintech markets | | | | | |
| The lack of digital literacy limits their use of Fintech for fear of losing the money they invest | | | | | |
| Not knowing the regulatory body that monitors Fintech generates doubt in using them | | | | | |
| Knowing the companies associated with the Fintech companies influences the use of these services | | | | | |
| The lack of regulation in Fintech is the main determinant for not using it | | | | | |
| Think Fintech companies are illicit | | | | | |
| Society has knowledge about guarantees and regulations concerning Fintech companies | | | | | |
| Financial education influences the decision to use Fintech for managing your finances | | | | | |
| Financial consumers have the appropriate financial advice to use Fintech | | | | | |
| Feel that limited financial education leads to less growth in the personal economy | | | | | |
| The lack of advice from banking entities is the main flaw in financial education | | | | | |
| That accessing Fintech may be useful influences my decision to use them | | | | | |
| Fintech companies facilitate the approval process and the requested loan amounts. | | | | | |
| Saving time when making transactions is motivating for using Fintech | | | | | |
| Offering low costs to use Fintech is a determining factor in deciding to use them | | | | | |

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
