# Peer review of "Adoption of Fintech Services in Young Students: Empirical Approach from a Developing Country"

_economies, doi:10.3390/economies11090226_

Round 1
Reviewer 1 Report
The paper titled “Adoption of fintech services in young students: empirical approach from a developing country”, focused on fintech adoption among university students in Columbia. The authors employed EFA and CFA approaches following the TAM model. Overall, I found the paper good but I hold the following observations:
Introduction section is well written, however, I think it can be further enriched by using latest articles on fintech adoption. There are several studies discussing the similar topic in several countries. in this context, I think the research gap and novelty should be of high priority and should be given more focus. The gap and how this study is novel should be given more attention. You can using the following researches to refine your research gap and present the contribution in a better way:
https://doi.org/10.3389/fpsyg.2022.984931
Research literature also need further refinement where we can see a critical literature review rather than just listing some studies and providing their results. Review of literature is more of literary debate which can provide the better argument how the research gap is presented and addressed. Add recent studies and rewrite the review part in more argumentative way.
Research methodology is tricky here, since the authors used 124 questionnaire which is not a healthy sample itself. I recommend using the above mentioned studies to improve your methodology part. In all of the studies, the sample size was considerably higher than 500 individuals. Secondly, provide table of how the sample size was distributed according to demographics.
Discussion and conclusion need further details. Specially the discussion where policy, practical and theoretical implication is missing. Add more research to support your findings.
I think the paper itself can be improved
Coherence needs to be improved. I observed many breaks where previous paragraph is not connected to the subsequent
Author Response
June 20, 2023
Dear
Economies – Editorial Team
Kind regards
In accordance with the suggestions of the reviewers in our article “Adoption of Fintech services among young students: An empirical approach in the Colombian context”, the following changes were made, properly marked with red letters in the article:
|
Reviewer |
Comment |
Response |
|
R1 |
Introduction section is well written, however, I think it can be further enriched by using latest articles on fintech adoption. There are several studies discussing the similar topic in several countries. in this context, I think the research gap and novelty should be of high priority and should be given more focus. The gap and how this study is novel should be given more attention. You can using the following researches to refine your research gap and present the contribution in a better way: https://doi.org/10.3389/fpsyg.2022.984931 |
Added recent studies on fintech adoption to the introduction to enrich it |
|
R1 |
Research literature also need further refinement where we can see a critical literature review rather than just listing some studies and providing their results. Review of literature is more of literary debate which can provide the better argument how the research gap is presented and addressed. Add recent studies and rewrite the review part in more argumentative way. |
Changes were made to the structure, adding recent studies to improve the literature corresponding to fintech adoption |
|
R1 |
Research methodology is tricky here, since the authors used 124 questionnaire which is not a healthy sample itself. I recommend using the above mentioned studies to improve your methodology part. In all of the studies, the sample size was considerably higher than 500 individuals. Secondly, provide table of how the sample size was distributed according to demographics. |
The sampling method used is detailed, taking into account that it is not a representative sample, but it is valid and useful for the purpose of this study, also arguing in the literature that the sample used 124 is an adequate number for something exploratory and not probabilistic. Also, it is attached as a limitation of this study that due to the sampling used, the results cannot be generalized. Table 1 of the article complements the results on the demographic information of the respondents with the distribution by sample size and not just percentage. |
|
R1 |
Discussion and conclusion need further details. Specially the discussion where policy, practical and theoretical implication is missing. Add more research to support your findings. |
The implications were rewritten taking into account the political, practical and theoretical aspects and more studies were placed in the discussion. |
|
R2 |
It is specifically recommended to develop the description of the literature review (as a separate part of article), |
The section was separated and the literary review was developed more in depth |
|
R2 |
It is specifically recommended to develop the description of of the managerial implications. |
The implications were rewritten taking managerial aspects into account. |
|
R3 |
Title, abstract and keywords. You should mention the country where the study is conducted. It will make the paper easy to find for readers. |
The country where the study is carried out was specified in the title, abstract and keywords |
|
R3 |
Introduction. The introduction is almost well-written. However, I encourage to enforce the real GAP and motivation in conducting this research paper. Also, a paragraph that would give a map of the paper. |
Emphasis is placed on the GAP in which it is intended to contribute to the study, including 4 references to support the need for this research. A paragraph is included with the guide of how the article is structured |
|
R3 |
1.1 Model and Proposed Hypotheses. More explanation of Figure 1 is needed. I can’t see why financial education would impact digital literacy and not vice versa. |
It is clarified that figure 1 summarizes and presents the proposed hypotheses. The possible relationship between financial education and digital literacy is supported through literature. Also, the explanation of the hypothesis is extended. |
|
R3 |
Methodology. This section is almost interesting. However, the authors need to elaborate on the sample size. Is the number enough to draw a general perception of the results? Also, in section 2.2 Instrument Design the authors did only cite two research papers. The authors need to expand the references in this section as well as clarify why they used interviews and why they used 3 parts. Also, a sample of the interview questions should be added to the appendix. |
The sampling method used is detailed, taking into account that it is not a representative sample but it is valid and useful for the purpose of this study, also arguing in the literature that the sample used 124 is an adequate number for something exploratory and not probabilistic. The division of the instrument was carried out only to have internal control and facilitate the understanding and analysis of the surveys in the database, this is also clarified in the methodology section as suggested. In section 2.2 Design of the instrument, the references are expanded to support the dimensions of the model. In the appendix, the complete survey is added, including the questions on the Liker scale that were used in the applied survey. |
|
R3 |
Results. The results section is interesting and almost well-written. However, can the authors explain why most of the samples were female? And also why most of the students were in the economic sciences. |
In the results, it is clear why most of the students are from the area of economic sciences. It is also made clear that in the universities that are the object of this study there is a representative number of women, also considering that a non-probabilistic convenience sampling was followed and the population to which the researchers had access was surveyed. |
|
R3 |
Discussion. This section looks strange to me. The subsections should be removed and a more cohesive discussion of the results should be done. |
Deleted subsections and connected paragraphs coherently |
We look forward to your comments and hope to hear from you soon.
Thank you very much
_
The authors

Reviewer 2 Report
In the assessment of the paper submitted for the review, I specifically focussed on the discussed issues, applied research procedure, substantive content of the paper and its structure.
The subject area raised in the paper is important. The reviewed paper is of scientific nature. The subject area discussed in the paper should be considered interesting. The considerations conducted in the paper are focused on such categories as: fintech, adoption models, developing countries, students.
The value of the paper results from combination of literature studies with the results of the empirical research. The research procedure has complex character. The structure of the paper is clear.
It is specifically recommended to develop the description:
- of the literature review (as a separate part of article),
- of the managerial implications.
Author Response

(The authors gave the same response as above.)

Reviewer 3 Report
Dear Authors,
Thank you for your analysis. Below are some constructive suggestions to strengthen your research paper.
Title, abstract and keywords
You should mention the country where the study is conducted. It will make the paper easy to find for readers.
Introduction
The introduction is almost well-written. However, I encourage to enforce the real GAP and motivation in conducting this research paper. Also, a paragraph that would give a map of the paper.
1.1 Model and Proposed Hypotheses
More explanation of Figure 1 is needed. I can’t see why financial education would impact digital literacy and not vice versa.
Methodology
This section is almost interesting. However, the authors need to elaborate on the sample size. Is the number enough to draw a general perception of the results? Also, in section 2.2 Instrument Design the authors did only cite two research papers. The authors need to expand the references in this section as well as clarify why they used interviews and why they used 3 parts. Also, a sample of the interview questions should be added to the appendix.
Results
The results section is interesting and almost well-written. However, can the authors explain why most of the samples were female? And also why most of the students were in the economic sciences.
Discussion
This section looks strange to me. The subsections should be removed and a more cohesive discussion of the results should be done.
Conclusion
This section is interesting and almost well-written.
There are some minor grammar and language mistakes that should be proofread.
Author Response

(The authors gave the same response as above.)

Round 2
Reviewer 1 Report
Dear Authors,
I do not hold any further observation on this manuscript. However, there are some issues with cited references, which i believe can be fixed in the step of final proofs before publication.
I wish you good luck